**Atmospheric Fe supply and marine productivity in the Glacial North Pacific Ocean**

François Burgay[1,2], Andrea Spolaor[1*], Jacopo Gabrieli[1], Giulio Cozzi[1], Clara Turetta[1], Paul Vallelonga[3,4], Carlo Barbante[1,2]

[1]Institute of Polar Sciences, National Research Council. Via Torino, 155, 3100 Venice (Italy)

[2]Department of Environmental Sciences, Informatics and Statistics, Ca' Foscari University of Venice. Via Torino, 155 – Venice (Italy)

[3]Physics of Ice Climate and Earth, Niels Bohr Institute, University of Copenhagen. Tagensvej 16, Copenhagen N2200 (Denmark)

[4]Oceans Graduate School, University of Western Australia (Australia)

     * Corresponding author: andrea.spolaor@unive.it

**Abstract**

Iron is a key element in the Earth climate system as it can enhance the marine primary productivity in the high-nutrient low-chlorophyll (HNLC) regions where, despite a high concentration of major nutrients, the chlorophyll production is low due to iron limitation. Aeolian mineral dust represents one of the main Fe sources to the oceans; thus, quantifying its variability over the last glacial cycle is crucial to evaluate its role in strengthening the biological carbon pump. Polar ice cores, which preserve detailed climate records in their stratigraphy, provide a sensitive and continuous archive for reconstructing past aeolian Fe fluxes. Here, we show the Northern Hemisphere Fe record retrieved from the NEEM ice core (Greenland), which offers a unique opportunity to reconstruct the past Fe fluxes in a portion of the Arctic over the last 108 kyr. Holocene Fe fluxes (0.042 -11.7 kyr b2k, 0.5 mg m$^{-2}$ yr$^{-1}$) at the NEEM site were four times lower than the average recorded over the last glacial period (11.7– 108 kyr b2k, 2.0 mg m$^{-2}$ yr$^{-1}$), while they were greater during the Last Glacial Maximum (LGM, 14.5 – 26.5 kyr b2k, 3.6 mg m$^{-2}$ yr$^{-1}$) and Marine Isotope Stage 4 (MIS 4, 60 - 71 kyr b2k, 5.8 mg m$^{-2}$ yr$^{-1}$). Comparing the NEEM Fe record with palaeoceanographic records retrieved from the HNLC

North Pacific, we found that the coldest periods, characterized by the highest Fe fluxes, were characterized by a low marine primary productivity in the subarctic Pacific Ocean, likely due to the greater sea-ice extent and the absence of major nutrients upwelling. This supports the hypothesis that Fe-fertilization during colder and dustier periods (i.e. LGM and MIS 4) was more effective in other regions, such as the mid-latitude North Pacific, where a closer relationship between marine productivity and the NEEM Fe fluxes was observed.

## 1. Introduction

Greenland and Antarctic ice cores are unique archives that can provide records of temperature, atmospheric dust load and atmospheric gas composition variability during the Holocene and the late Pleistocene (Jouzel et al., 1996; Lambert et al., 2008; Schüpbach et al., 2018; Watanabe et al., 2003). Glacial periods were dustier and characterized by a lower $CO_2$ concentration ($\approx 180$ ppm) than interglacials ($\approx 280$ ppm) (Lambert et al., 2008; Lüthi et al., 2008). This dichotomy is explained through several different hypotheses: the increase in aridity and newly exposed continental shelves (Fuhrer et al., 1999), an increase in the aerosol atmospheric life-time resulted from a reduced hydrological cycle (Lambert et al., 2008; Yung et al., 1996), increased glacial-derived mobilization of highly bioavailable iron (Fe) from physical breakdown of bedrock (Shoenfelt et al., 2018), and, lastly, more vigorous polar circulation capable of entraining additional dust from lower latitudes (Mayewski et al., 1994). Regardless of the source, the higher atmospheric burden of mineral dust during glacial periods affected climate through both physical and biological mechanisms. Dust particles can directly influence the Earth's radiative budget by scattering, absorbing and re-emitting shortwave and longwave radiation (Miller and Tegen, 1998; Schepanski, 2018). During the LGM, model results showed that the enhanced dust transport caused, alone, a 1.0 W/m² globally averaged radiative forcing decrease compared to present day conditions, which contributed to a 0.85°C cooling relative to the current climate (Mahowald et al., 2006). Conversely, once deposited on the ocean surface, the mineral dust delivered major and micronutrients (including Fe) that could have stimulated the biological carbon pump (Martin et al., 1990). Indeed, Fe can limit marine primary production (MPP) in the high-nutrient low-chlorophyll (HNLC) oceans, which are characterized by a high concentration of nutrients, but low productivity (Martin et al., 1990). The largest ones are the Southern Ocean, the Equatorial Pacific and the North Pacific Ocean (Duggen et al., 2010). In these regions, the Fe role in modulating marine productivity was demonstrated through both artificial Fe fertilization experiments (Smetacek et al., 2012; Tsuda et al., 2003; Yoon et al., 2018) and natural Fe inputs from iceberg

melting, volcanic eruptions and glacially sourced dust (Duprat et al., 2016; Langmann et al., 2010; Shoenfelt et al., 2017). For its biological relevance, it has been hypothesized that the recorded decrease in the atmospheric $CO_2$ concentration during glacial periods was linked to the Fe-modulated enhancement of the biological carbon pump in the HNLC regions due to the increase in Fe availability (Martin et al., 1990). Evidences for the existence of a strong link between atmospheric Fe deposition and marine productivity were retrieved from a marine sediment core collected in the subantarctic zone of the Southern Ocean where, the coldest periods were mirrored by an increase in atmospheric Fe fluxes and by an enhancement of both MPP and degree of nutrient consumption (Martínez-García et al., 2014). Yet, according to both modeling (Lambert et al., 2015) and observational (Gaspari et al., 2006; Röthlisberger, 2004; Vallelonga et al., 2013) studies, the Fe-fertilization mechanism itself cannot completely explain the $\approx$100 ppmv glacial-interglacial atmospheric $CO_2$ variability, but only around 8-20 ppmv of it (Lambert et al., 2015).

However, the role of Fe fertilization in the Northern Hemisphere and in the HNLC region of the North Pacific is unclear due to the few available Arctic Fe flux records which are either limited to the last century or they only cover short time periods (Burgay et al., 2019; Hiscock et al., 2013). Thus, reconstructing how the Fe concentrations and fluxes have changed in the Northern Hemisphere during the last glacial cycle is essential to understand the evolution of the global atmospheric circulation, the human impact on dust mobilization (Mahowald et al., 2008) and to evaluate, as well, the impact that Fe might have had on MPP in the North Pacific HNLC region. Here, we present a high-resolution 108 kyr record of total dissolvable Fe (TDFe) retrieved from the North Greenland Eemian Ice Drilling (NEEM) ice core (Rasmussen et al., 2013; Schüpbach et al., 2018), which provides a unique insight on the atmospheric Fe supply in the Arctic both during the Holocene and the last glacial period. Furthermore, we performed a comparison between the TDFe NEEM record and various palaeoproductivity records from the HNLC North Pacific region (Figure 1) to evaluate whether the increase in aeolian Fe fluxes was mirrored by an increase in marine productivity. We underline that, TDFe concentrations, as it will be discussed in the following, derives from the acidification of the snow samples for 1 month at pH 1. Thus, they represent an upper limit of the aeolian Fe potentially available for the phytoplankton, and it might overestimate the actual bioavailable Fe.

## 2. Materials and methods

## 2.1 Sampling and processing

In the framework of the NEEM project, a 2540 m-depth ice core was drilled in north-western Greenland (77°45'N, 51°06'W) at 2479 m.a.s.l. The site is characterized by an average annual temperature of -29°C and a modern accumulation of 22 cm ice equivalent per year. According to the GICC05modelext-NEEM-1 timescale, the ice core covers the last 128 kyrs (Rasmussen et al., 2013). The ice cores were cut to obtain ice sticks with a square cross section of 36x36 mm. They were continuously melted on a continuous flow analysis (CFA) system with a typical melt-speed of 3.5 cm min$^{-1}$ (Schüpbach et al., 2018). The CFA system provides meltwater from the inner and least likely to be contaminated part of the core, thus we did not adopt any further decontamination procedure. The ICP-MS samples were manually collected at a low-resolution (110 cm). The temporal resolution depends on the accumulation rate and it decreases with depth because of the ice thinning. According to the available timescale (Rasmussen et al., 2013) and considering the 110 cm sampling resolution, the temporal resolution varies from decadal to millennial (Table 1).

Samples were collected in vials previously cleaned as follows: 7 days with $HNO_3$ 5% (Suprapure, Romil, UK), rinsed three times with ultrapure water (UPW, Elga, UK), 7 days with $HNO_3$ 2% (Suprapure, Romil, UK), rinsed three times with UPW and then stored in $HNO_3$ 1% (Ultrapure, Romil, UK) until the day before the sample collection, when they were rinsed three times with UPW and dried overnight under a laminar flow hood Class 100. The samples were kept frozen and shipped to Italy for analysis. Once melted, the samples were acidified to pH 1 using $HNO_3$ (Suprapure, Romil, UK). To ensure an effective dissolution of Fe particles, samples were stored at room temperature and analysed 30 days after the acidification without any additional filtration step. We adopted this approach since the analysis immediately after the acidification step might have led to uncertainties attributable to the Fe dissolution kinetics (Edwards, 1999; Koffman et al., 2014). Our choice was consistent with other studies that indicate that samples to be used for calculation of atmospheric fluxes must be acidified for at least 1 month prior to analysis to avoid any possible misinterpretation of the trace-element data (Koffman et al., 2014). We will refer to this fraction as total dissolvable Fe (TDFe) which includes both the most labile fraction (dissolved iron, DFe), which is rendered soluble under mildly acidic conditions (Hiscock et al., 2013), and the fraction enclosed in iron-bearing mineral particles. TDFe does not directly represent the actual bioavailable Fe that can be dissolved into seawater at pH 8, but, considering that

TDFe and DFe are significantly correlated (Du et al., 2020; Xiao et al., 2020),  an upper limit of the aeolian
Fe potentially  available for the phytoplankton (Edwards et al., 2006).
**2.2 Analytical procedure and performances**
The ice samples were analysed with an Inductively  Coupled Plasma Single  Quadrupole Mass Spectrometer
(ICP-qMS, Agilent 7500 series, USA) equipped with a quartz Scott spray chamber for the determination of
Ca, Na and Fe. To minimize  any kind of contamination,  all the instrument  tubes were flushed before the
analysis for 2 hours with 2% $HNO_3$ (Suprapure, Romil, UK). A 120 seconds rinsing  step with 2% $HNO_3$
(Suprapure, Romil, UK) occurred after each sample analysis to reduce any possible memory effect. The vials
used for the standard preparation were cleaned following the same procedure adopted for the ice samples.
Considering  the isobaric and polyatomic  interferences affecting Fe, this element  was quantified  using the
interference-free isotope $^{57}Fe$. External calibration  curves with acidified standards (2% $HNO_3$, Suprapure,
Romil, UK) were prepared for Ca, Na and Fe from dilution  of a certified single-element  1000 ppm $\pm$ 1%
standard solution (Fisher Chemical, USA). The resulting  $R^2$ for the external calibration  curves was 0.999 for
all the elements. The Limit of Detection (LoD) for $^{57}Fe$, calculated as three times the standard deviation of the
blank,  was 0.8 $\mu g$ $L^{-1}$. To assess accuracy for Fe, the TM-RAIN04 certified reference material (National
Research Council of Canada) was measured every 50 samples. The accuracy was determined as a recovery
percentage calculated as O/T %, where O is the determined value and T is the certified value. For Fe, the
accuracy was 104%, while precision,  calculated as Relative Standard Deviation (RSD %) of selected samples
read multiple  times (n = 5) during the analysis, was on average 5% (7% for samples (n = 3) from the interglacial
period, 4% for samples (n = 3) from the last glacial period). For Ca and Na, the LoD was 1 $\mu g$ $L^{-1}$ and 3 $\mu g$ $L^{-}$
$^1$, respectively.  In the absence of a certified reference material, Ca and Na accuracy was calculated using a
Quality Control (QC) sample prepared at 10 $\mu g$ $L^{-1}$ and measured every 50 samples. Accuracy for Ca and Na,
calculated as described above, was 94% and 108%, respectively, while precision (RSD%) was on average 6%
(4% for samples from the interglacial period and 7% for samples from the last glacial period) and 2% (for both
periods), respectively.
Non-sea-salt Ca concentration is commonly  used as proxy for terrestrial inputs in polar regions and it is
calculated as nssCa = [Ca] - ([Ca]/[Na])$_{sw}$ $\cdot$ [Na], where sw indicates seawater.

## 3. Results and discussion

### 3.1 Fe fluxes from the NEEM core

Fe and nssCa concentrations and fluxes were calculated as $F = C \cdot A$ (where F is the Fe flux, in mg m$^{-2}$ yr$^{-1}$, C is the Fe or nssCa concentration, in ng g$^{-1}$, and A the accumulation, in m yr$^{-1}$ ice equivalent, previously determined by Rasmussen et al., 2013). A pattern of higher dust (expressed as nssCa$^{2+}$) and Fe fluxes during colder climate periods and lower dust and Fe fluxes during warmer climate periods is clearly recognizable (Figure 2).

The Holocene (0.042 -11.7 kyr b2k) was characterized by average Fe fluxes of 0.5 mg m$^{-2}$ yr$^{-1}$ that varied between 0.01 mg m$^{-2}$ yr$^{-1}$ and 5.3 mg m$^{-2}$ yr$^{-1}$ (Figure 2). The coefficient of variability (CV), calculated as the ratio between the standard deviation and the mean value, was 1.2. The more recent 4000 years are characterized by the highest average Fe fluxes (0.6 ± 0.4 mg m$^{-2}$ yr$^{-1}$). The lowest Fe fluxes were recorded between 4000 and 8000 years b2k (0.3 ± 0.2 mg m$^{-2}$ yr$^{-1}$). During the Younger Dryas (YD, 11.7 − 12.9 kyr b2k), an abrupt cooling was observed with a drop in the δ$^{18}$O value from -36.9‰ to -43.1‰. Coincidently, the recorded average Fe fluxes rose to 1.2 ± 0.4 mg m$^{-2}$ yr$^{-1}$, higher than both the 12.9-13.9 kyr b2k (0.5 ± 0.3 mg m$^{-2}$ yr$^{-1}$) and the 10.7- 11.7 kyr b2k (0.3 ± 0.2 mg m$^{-2}$ yr$^{-1}$) periods.

The last glacial period (11.7-108 kyr b2k) showed Fe fluxes four-times higher (2.0 ± 2.2 mg m$^{-2}$ yr$^{-1}$) than the Holocene, spanning from 0.05 to 16.5 mg m$^{-2}$ yr$^{-1}$ (Figure 2). However, a significant variability during the last glacial period was detected. During the LGM and MIS 4, average Fe fluxes were 7 (3.6 ± 2.3 mg m$^{-2}$ yr$^{-1}$) and 10-times (5.8 ± 2.8 mg m$^{-2}$ yr$^{-1}$) greater than the Holocene average. Fe fluxes also increased during the MIS 5c-MIS5b transition (87 kyr b2k), when a concurrent decrease in δ$^{18}$O values was observed. During MIS 5c and MIS 5d, Fe fluxes were comparable with those detected during the Holocene. The high frequency of the Dansgaard-Oeschger (D-O) events during MIS 3, was mirrored by the high variability in both nssCa and Fe fluxes. Each stadial period corresponded to an increase in both Fe and nssCa. However, their variability was significantly different. During MIS 3, Fe fluxes showed maximum values greater than 5 mg m$^{-2}$ yr$^{-1}$ during D-O 4, 9, 12, 15 (8.5, 6.5, 7.5, 6.6 mg m$^{-2}$ yr$^{-1}$ respectively), and lower than 5 mg m$^{-2}$ yr$^{-1}$ during D-O 6, 7, 8, 10, 11 and 13 (3.9, 2.6, 4.1, 2.6, 2.7, 3.2 mg m$^{-2}$ yr$^{-1}$ respectively). This variability was significantly higher than the one recorded for nssCa, which showed maximum values close to 20 mg m$^{-2}$ yr$^{-1}$ for all the D-O events.

## 3.2 Comparison with Fe fluxes from Antarctic ice cores

The NEEM Fe ice core record allows the first comparison of Fe concentrations and fluxes between the Arctic and Antarctica (Figure 3, Table 3). The only Antarctic Fe records that can reach at least the LGM are from Talos Dome (TD) (Spolaor et al., 2013; Vallelonga et al., 2013), Law Dome (LD) (Edwards et al., 2006; Edwards et al., 1998) and EPICA Dome C (EDC) (Wolff et al., 2006). However, we point out that both the samples from Dome C and Talos Dome were acidified for at least 24 hours, leading to a possible underestimation of the actual TDFe concentration. This implies that the general trends and features can be comparable with the NEEM record, while absolute concentrations might differ due to the different acidification procedure used (Koffman et al., 2014).

During the Holocene, in Antarctica, the average Fe flux and concentration values varied significantly among the different sites with similar values recorded at the coastal sites (TD) and lower values in the internal Antarctic Plateau (EDC) (Table 3). For TD, this was explained both through changes in atmospheric transport patterns across Antarctica and through an additional local input of dust from proximal Antarctic ice-free zones that affected coastal sites more than the central plateau, which was exclusively exposed to remote sources such as southern South America (Albani et al., 2012; Delmonte et al., 2010b; Vallelonga et al., 2013).

During the LGM, both TD and EDC shared a similar dust flux loading, comprised between 10 and 15 mg $m^{-2}$ $yr^{-1}$ (Baccolo et al., 2018), and the same dust source region, as confirmed by the Sr-Nd isotopes (Delmonte et al., 2010a). Compared to the Holocene, in TD the atmospheric dust fluxes increased of a factor 6, while in EDC the increase was approximately of a factor 25 (Delmonte et al., 2010b). This is mirrored by a similar average Fe fluxes enhancement compared to the Holocene with values that were up to 4 and 21-fold higher, respectively (Vallelonga et al., 2013; Wolff et al., 2006). The reason of these discrepancies between the two sites is likely due to the higher Holocene dust flux observed in TD compared to EDC, as a consequence of a relevant local dust contribution at TD (Baccolo et al., 2018; Delmonte et al., 2010b).

During the last glacial period, the most relevant dust source was southern South America for both TD and EDC (Basile et al., 1997; Delmonte et al., 2010b; Lambert et al., 2008). Dust fluxes peaked during MIS 4 where both sites recorded maximum values around 10 mg $m^{-2}$ $yr^{-1}$ (Lambert et al., 2008; Vallelonga et al.,

2013) and comparable Fe fluxes ($0.17 \pm 0.07$ mg m$^{-2}$ yr$^{-1}$ at TD and $0.12 \pm 0.07$ mg m$^{-2}$ yr$^{-1}$ at EDC) (Vallelonga
et al., 2013; Wolff et al., 2006).
The LD record, due to the different analytical preparation of the samples, is not directly comparable with
TD and EDC. Nevertheless, we can still evaluate and discuss the Fe flux ratio between the Holocene and the
LGM. Unfortunately, for the LD record, there is no dust profile available, meaning that it is not possible to
assess which is the main dust and Fe sources for this location, although the Australian continent has been an
important source of mineral dust in the recent past (Edwards et al., 2006; Vallelonga et al., 2002). During the
LGM, Fe fluxes increased 10-fold compared to the Holocene period, 2.5 times more than what was observed
in TD. Similarly to what observed in the EDC record, this difference might be explained either by the absence
of local dust sources that affected LD during the Holocene, or by the lower sampling frequency for the LD
record (n = 27) compared to TD (n = 801).
Despite the different acidification times, the overall picture during the Holocene is that the average Fe
fluxes in NEEM (0.5 mg m$^{-2}$ yr$^{-1}$, CV = 1.2) were higher than in Antarctica. Among the Antarctic Fe fluxes,
TD (0.09 mg m$^{-2}$ yr$^{-1}$, CV = 1.2) and LD (0.04 mg m$^{-2}$ yr$^{-1}$, CV = 0.5) were higher than the ones recorded at
EDC (0.007 mg m$^{-2}$ yr$^{-1}$, CV = 0.2).
In NEEM, the LGM (19 – 26.5 kyr b2k) was characterized by a 10-fold and 7-fold enhancement in dust
(expressed as nssCa) and Fe fluxes, respectively. A similar behaviour was observed in the Antarctic cores as
described above (Table 3). Considering that the atmospheric $CO_2$ concentration dropped down to 180 ppm
(Köhler et al., 2017), the global Fe fluxes enhancement likely contributed to part of this decrease, promoting
marine productivity in some HNLC regions (Amo and Minagawa, 2003; Kawahata et al., 2000; Martínez-
Garcia et al., 2011).
During MIS 4 (60-71 kyr b2k), NEEM Fe fluxes were higher compared to all the other investigated
records. Compared to the LGM average, during MIS 4, dust (Ruth, 2007), nssCa and Fe fluxes (this work) in
the Arctic exhibited a $\approx 1.5$-fold increase (Table 3), while they were lower both in TD and EDC. To explain
this behaviour we advance some hypotheses. The first is that the increase in dust and Fe fluxes can be
attributable to changes in the atmospheric circulation, likely due to the topographic influence of the Laurentide
Ice Sheet (LIS). Indeed, during the LGM, LIS was nearly 2 times larger than at MIS 4 (Löfverström et al.,
2014; Tulenko et al., 2020) and it might have caused a stronger meridional splitting of the westerlies
(Löfverström et al., 2014) and a southward migration of their mean position (Kang et al., 2015; Manabe and
Broccoli, 1985). The southward shift during the LGM might have produced a reduction of strong winds passing
over the source areas (i.e. Taklimakan and Gobi deserts) (Kang et al., 2015) and/or a stronger southward Fe
and dust deposition over the Chinese Loess Plateau (Zhang et al., 2014) and the mid-latitude North Pacific
(Sun et al., 2018). In contrast, during MIS 4, the westerlies might have been located northward (i.e. over the
Taklimakan and Gobi deserts) and characterized by a less marked meridional splitting (Löfverström et al.,
2014), conveying a larger amount of dust to Greenland. We also propose two alternative hypotheses that rely
on 1) the possibility that additional dust sources (e.g. Saharan dust) might have reached Greenland during MIS
4, and 2) that during MIS 4, the Asian monsoon system was stronger in winter than in summer, producing drier
conditions that caused an enhanced dust production and transport to Greenland (Xiao et al., 1999). However,
to better address this point, a more comprehensive investigation that involves a large set of palaeorecords and
atmospheric modelling is required and it is beyond the scope of the manuscript.
**3.3 Comparison with lower-resolution Fe NEEM measurements**
A parallel study that reported Fe concentration from the NEEM ice core was recently published (Xiao et
al., 2020). It reports the TDFe and DFe concentration and fluxes with a lower temporal resolution (n = 166)
than the current investigation (n = 1596). Moreover, the analytical approach was different since the melted ice
samples were filtered at 0.45 μm and acidified for six weeks before the analysis. Even though the overall
pattern between the two records is similar, we observe several differences between Xiao et al., (2020) and our
study: a) the average Fe concentration over the entire record is 4-fold higher than the one found in our
investigation (101.4 ng $g^{-1}$ vs 20.4 ng $g^{-1}$); b) the Fe concentration range is wider (1.5-1194.5 ng $g^{-1}$ vs >LoD -
457.6 ng $g^{-1}$) compared to the data presented in this manuscript; c) average Fe fluxes are 2.4-fold higher during
the Holocene (1.2 mg $m^{-2}$ $yr^{-1}$ vs 0.5 mg $m^{-2}$ $yr^{-1}$) and 3.5-fold higher during the LGM (12.5 mg $m^{-2}$ $yr^{-1}$ vs 3.6
mg $m^{-2}$ $yr^{-1}$) that the ones recorded in this study; d) the LGM Fe flux showed a 10-fold increase during the
Holocene, compared to the 7-fold enhancement that we observed; e) TDFe fluxes and concentration are higher
during the LGM than during MIS 4, while we found higher fluxes during MIS 4, consistently with a similar
enhancement of nssCa and dust (Ruth, 2007).
Possible reasons of these differences might rely on the different temporal resolution and on the
discrepancies between the adopted analytical approaches that highlight the need to standardize the analytical
procedures when trace elements are analyzed in ice and snow samples in order to have a more reliable
comparison among both different and identical locations.
**3.4 Fe and marine productivity in the Northern Hemisphere**
Considering the biological relevance of Fe and taking advantage from the Fe flux record retrieved from
the NEEM ice core, one important question remains regarding whether its flux increase during the last glacial
period triggered the marine productivity in the HNLC region of the North Pacific (Olgun et al., 2011).
Nowadays, a significant amount of Asian dust (250 Mt $yr^{-1}$) is primarily deposited over the HNLC region
of the subarctic Pacific (Serno et al., 2014; Zhang et al., 2003) and the marine productivity changes in this
oceanic region might reflect potential Fe fertilization effects promoted by atmospheric Fe supply. During
modern times, both increases in aeolian influx from Asia (Young et al., 1991) and sporadic Fe input from
volcanic eruptions (Langmann et al., 2010) resulted in enhanced MPP by more than 60%. Moreover, recent
Fe-fertilization experiments performed south of the Gulf of Alaska (McDonald et al., 1999; Tsuda et al., 2003),
showed significant increases in the abundance of diatoms and in chlorophyll-a concentration (Boyd et al.,
1996), indicating that the North Pacific is maybe sensitive to atmospheric Fe inputs. However, no data are
available to evaluate if the Fe-sensitivity of the subarctic Pacific Ocean holds even over longer timescales and,
if an increase in the aeolian Fe supply, observed during glacial periods, could explain the MPP variability in
the subarctic Pacific Ocean. To address this point, we compared the NEEM Fe record with different marine
sediment cores (Table 4).
Previous geochemical evidence indicates that for both interglacial and glacial periods the dust source
influencing Greenland and the North Pacific mainly originated from the East Asian deserts (Schüpbach et al.,
2018; Serno et al., 2014). However, considering that there are no aeolian Fe flux records from the marine
sediment cores, they might have received different amount of Fe compared to what observed in the ice core
record. Through a comparison between a marine sediment record from the western subarctic Pacific Ocean
(SO202-07-6) and the NGRIP ice core, it has been shown that dust fluxes changed coherently and
simultaneously during abrupt climate changes, even though with different amplitude (Serno et al., 2015). The
larger variability observed in NGRIP, as well as in NEEM, than in marine sediments, indicates changes in the
atmospheric dust transport from the source areas to Greenland (e.g. rate of aerosol rainout, different wind
strength).

Recently, it has been proposed that additional dust sources might have influenced Greenland in the last

31 kyrs (Han et al., 2018; Lupker et al., 2010). Strontium and lead isotopes indicate that Saharan dust
contributed to the overall NEEM dust budget during the Younger Dyras (12-73%) and between 17 kyrs and
22 kyrs (16-70%), while the Taklimakan and Gobi contribution (i.e. eastern Asia sources) was dominant (55-
94%) prior to 22 kyrs (Han et al., 2018). However, despite the Saharan dust source, we assume that the main
dust source for the NEEM ice core during the last glacial period is represented by the Gobi and Taklimakan
deserts (Svensson et al., 2000). This is coherent with the dust changes synchronicity among Greenland, the
Chinese loess (Ruth et al., 2007) and the Northern Pacific sediment records located downwind of the Asian
dust sources (Schüpbach et al., 2018; Serno et al., 2015). Nevertheless, additional investigations to assess the
magnitude of the Saharan dust contribution prior to 31 kyrs and to identify other possible source regions are
needed.

All variables considered (i.e. different dust amplitude and other potential dust sources), and observing

that the overall pattern of higher dust deposition during the coldest periods is consistent between the ice and
sediment core records, we assumed that the Fe flux changes observed in NEEM are representative for the
aeolian Fe supply to the subarctic Pacific Ocean.

To evaluate whether past marine productivity was influenced by atmospheric Fe supply for the period

ranging from the LGM to the Holocene, we compared the NEEM record with the high temporal resolution
SO202-27-6 (from the Patton-Murray Rise plateau, eastern subarctic Pacific Ocean) and the SO202-07-6 (from
the Detroit Seamount, western subarctic Pacific Ocean) productivity records (Méheust et al., 2018). For a long-
term record, we relied on the ODP887 (McDonald et al., 1999) and the ODP882 (Haug et al., 1995) sediment
cores, located close to SO202-27-6 and SO202-07-6, respectively. A comparison over the last 108 kyr between
the NEEM record and the S-2 sediment core (from the Shatsky Rise, mid-latitude North Pacific) was also
performed (Amo and Minagawa, 2003) (Figure 4, Table 4).
The past marine primary productivity reconstruction was performed relying on the Si/Al ratio, % of
biogenic silica and brassicasterol concentration. Si/Al ratio is used as a proxy for opal, or biogenic silica
(diatoms), in the absence of directly measured opal concentrations. The normalization to Al removes any
possible variable inputs of lithogenic detritus (McDonald et al., 1999). Brassicasterol is a sterol compound
which has been used as a molecular indicator of the presence of diatoms (Sachs and Anderson, 2005).
Brassicasterol concentration is also used, together with highly branched isoprenoid alkenes ($IP_{25}$), for the $PIP_{25}$
calculation, which is a proxy for the evaluation of past sea-ice conditions (Méheust et al., 2018; Müller et al.,

2011)

### 3.4.1 From the LGM to the Holocene

During the Last Glacial Maximum, the Fe fluxes recorded in the NEEM ice core were 7 times higher
compared to the Holocene. However, marine productivity in the subarctic Pacific Ocean, expressed as Si/Al
ratio (McDonald et al., 1999), % biogenic silica (Haug et al., 1995) and brassicasterol concentration (Méheust
et al., 2018), was at its lowest level (Figures 4, 5). Reconstructions based on the foraminifera-bound $\delta^{15}N$ (FB-
$\delta^{15}N$), a proxy which indicates the degree of nitrate consumption by phytoplankton (Martínez-García et al.,
2014), showed that, in the western subarctic Pacific Ocean, the nitrate consumption was more complete during
the LGM and the YD (i.e. when MPP was low) compared to the warmest periods (Ren et al., 2015). In other
words, during the coldest and dustiest periods, the nitrate consumption efficiency was higher (i.e. increase in
the FB-$\delta^{15}N$ values) than during the interglacials, even though MPP was low. This apparent contradiction can
be explained by an increase in water stratification (either by reduced upwelling or vertical mixing), where the
most nutrient-rich and oxygen depleted waters were shifted to deeper depths, while nutrient-depleted and
better-ventilated waters rested above a hydrographic boundary at 1500-2000 m (Kohfeld and Chase, 2017).
Water stratification led to minimal input of nutrients to the surface ocean, leading the system towards a major
nutrient limitation (Kienast et al., 2004; Ren et al., 2015). Among the several possible reasons that can explain
the increase in water stratification in the Glacial North Pacific, we propose two hypotheses. The first relies on
the glacial closure of the Bering Strait that reduced the freshwater export from the Pacific Ocean to the Atlantic,
retaining more freshwater in the North Pacific (Talley, 2008). The second involves sea-ice formation. When
sea-ice forms, in the Okhotsk and Bering Seas, brine rejection occurs, increasing water density and creating
the more saline and denser North Pacific Intermediate Water (NPIW). When the wind blows the sea-ice away

from where it was originally formed, brine rejection can further proceed at the same location following the formation of new sea-ice. The continuous brine rejection promotes the freshening of surface waters and strengthens water stratification (Costa et al., 2018).

An additional explanation for the observed lower productivity during glacial periods arises from the higher extent of perennial sea-ice that might have played a role in creating a physical barrier between the atmosphere and the marine environment, reducing the amount of available sunlight and the direct deposition of bioavailable Fe on the seawater surface (Kienast et al., 2004; Méheust et al., 2018). Marine sediment records, collected in the eastern and western subarctic Pacific and in the Bering Sea, showed extended spring ice-cover during the LGM (Méheust et al., 2018; Méheust et al., 2016) when the Fe fluxes were at their maxima. The progressive decrease in perennial sea-ice coverage recorded after the LGM led to an increase in the marine productivity (Figure 5), with a maximum during the Bølling-Allerød (B/A) warm event ($\approx$ 13-15 kyr ago). The possible relevance of sea-ice in modulating MPP at the highest latitude of the Pacific Ocean during the LGM is strengthened by a marine sediment record collected in the mid-latitude North Pacific (Amo and Minagawa, 2003), which, because of its southernmost location, did not experience any sea-ice condition. During the LGM, contrarily to what is observed in the subarctic Pacific, a prominent maximum in marine productivity was recorded, suggesting that Fe could have triggered an important phytoplankton response (Figure 4d). The Fe-sensitivity of the mid-latitude North Pacific is confirmed during the Holocene, when the Fe fluxes were at their minima and the productivity, expressed as MAR (mass accumulation rate) $C_{37}$ alkenone ($\mu g\ cm^{-2}\ kyr^{-1}$), was at its lowest level. A plausible explanation is that stratified waters did not characterize this region during the last glacial period and thus it was not affected by the limitation of major nutrients. Unfortunately, neither FB-$\delta^{15}N$ nor information about water stratification are available for this record.

However, there might be other reasons that could explain the strengthening in MPP during the B/A warm period in the subarctic North Pacific. Among them, we propose the increase in the sea-level that inundated previously exposed lands which might have entrained iron and other nutrients to the marine ecosystem (Davies et al., 2011), or changes in the oceanic circulation (McManus et al., 2004). Indeed, at the onset of the B/A event, the meridional overturning circulation rapidly accelerated and this might have produced an upward displacement of the nutrient-rich North Pacific Deep Water towards intermediate depths, promoting an injection of nutrients to surface waters and enhancing marine productivity.

These additional explanations shed light on the marginal role that atmospheric Fe fertilization had in

promoting MPP in the subarctic Pacific Ocean since other players might have had a more significant impact
(Kohfeld and Chase, 2017).
**3.4.2 From 108 kyr to the LGM**

According to the available records, marine productivity changed heterogeneously in the Pacific Ocean

during the last glacial period (Figure 4).

It is challenging to state, with a high degree of confidence, whether Fe-fertilization triggered a

phytoplankton bloom or not in the HNLC subarctic North Pacific. This is due to the different responses that
the western and the eastern side of the subarctic North Pacific showed with respect to the atmospheric Fe
supply (Figure 4). In the eastern subarctic Pacific, the increase in the aeolian Fe fluxes was mirrored by a
phytoplankton response during the MIS 5.2 and the MIS 5 / MIS 4 transition. The subsequent decrease in MPP
during the MIS 4 suggests that the prolonged Fe supply during the coldest stadial might have led the ecosystem
towards the limitation of other nutrients (Kienast et al., 2014) following the same mechanisms described in the
previous section. The enhanced water stratification during those periods, as suggested by stable oxygen isotope
ratios in planktonic foraminifera (Zahn et al., 1991), did not allow a supply of macronutrients from below the
mixed layer. Thus, additional atmospheric Fe supply had little effect on phytoplankton productivity, suggesting
their growth was likely limited by the lack of major nutrients (Kienast et al., 2004). In the western subarctic
Pacific, the increase in productivity was recorded also in periods with low atmospheric Fe fluxes (e.g. from
100 to 90 kyr at ODP882), strengthening the hypothesis that other influences (e.g. meltwater inputs, continental
margin supply, sea-ice) had a more relevant role (Kienast et al., 2004; Lam and Bishop, 2008) than atmospheric
Fe supply.

On the contrary to what was observed in the subarctic Pacific, the S-2 sediment core collected in the

mid-latitude North Pacific (Amo and Minagawa, 2003), showed a marked increase in marine primary
productivity during MIS 4 and the overall last glacial period when the Fe fluxes were higher (Figure 4). MPP
in the mid-latitude North Pacific might have been more sensitive to the atmospheric Fe supply, suggesting that
the high degree of upper ocean stratification that characterized the subarctic region of the Pacific Ocean did
not affect the mid-latitude North Pacific allowing for a continuous supply of macronutrients. The observed
increase in dust transport (and Fe deposition) could have then stimulated marine productivity (Kienast et al.,

2004).

## 4. Conclusions and future perspectives

In this study, we provided a high-temporal-resolution Fe record from mineral dust input retrieved from the
NEEM ice core. Through the comparison with other available Fe records, we observed that Fe fluxes were
higher in Greenland than in Antarctica. The greatest difference between Arctic and Antarctic records occurred
during MIS4, when Fe fluxes in NEEM were 1.5 times higher than during the LGM, while, in TD and EDC,
they were lower. To explain this behaviour, we advanced two hypotheses (i.e. change in the atmospheric
circulation or additional dust sources that reached Greenland), even though more detailed investigations are
needed.
Merging our record with marine productivity data, we found that a link between Fe transport and ocean
productivity holds in the mid-latitude North Pacific, indicating that this area might be sensitive to the
atmospheric Fe supply. On the contrary, in the subarctic Pacific, we did not find any overwhelming evidence
that the increase in the atmospheric Fe fluxes triggered a phytoplankton response. This indicates that other
players, such as sea-ice and increased water stratification during the coldest periods had a more relevant role
in modulating the MPP in the HNLC region of the North Pacific on a millennial time scale.
This study provides an upper limit for estimating the potentially bioavailable Fe supplied to marine
phytoplankton in the North Pacific region, however additional studies should focus on analysing the labile and
bioavailable Fe fractions to constrain realistic Fe supply and response of the marine ecosystem.
**Data availability**
Data will be published on Pangaea
**Author contributions**
FB wrote the manuscript. FB, AS and CB designed the research. JG, CT and GC performed the analyses. PV
contributed to the interpretation of the results.
**Competing interests**
The authors declare that they have no conflict of interest.

## Acknowledgments

We sincerely thank all the persons involved in the logistics, drilling operations, ice-core processing and sample
collection. NEEM is directed and organized by the Center of Ice and Climate at the Niels Bohr Institute and
US NSF Office of Polar Programs and it is supported by funding agencies and institutions in Belgium (FNRS-
CFB and FWO), Canada (NRCan/GSC), China (CAS), Denmark (FIST), France (IPEV, CNRS/INSU, CEA
and ANR), Germany (AWI), Iceland (RannIs), Japan (NIPR), Korea (KOPRI), The Netherlands (NWO/ALW),
Sweden (VR), Switzerland (SNF), United Kingdom (NERC), and the USA (US NSF, Office of Polar
Programs).
We are grateful to the three anonymous reviewers and to the editor that contributed to the overall improvement
of the manuscript.

**Figures and tables**

**Figure 1 -** Locations of the NEEM ice core (blue diamond, this study), the LD ice core (pink triangle, Edwards
et al., 2006), EDC ice core (black diamond, Wolff et al., 2006) and TD ice core (green diamond, Vallelonga
et al., 2013). We retrieved palaeoproductivity data for the eastern North Pacific (black triangle) from the
ODP882 (Haug et al., 1995) and SO202-27-6 (Méheust et al., 2018) sediments cores, while for the western
Pacific Ocean (red triangle) from the ODP887 (McDonald et al., 1999) and SO202-07-6 (Méheust et al., 2018)
sediment cores. The palaeoproductivity record from the mid-latitude North Pacific was retrieved from the S-2
sediment core (blue triangle, Amo and Minagawa, 2003).



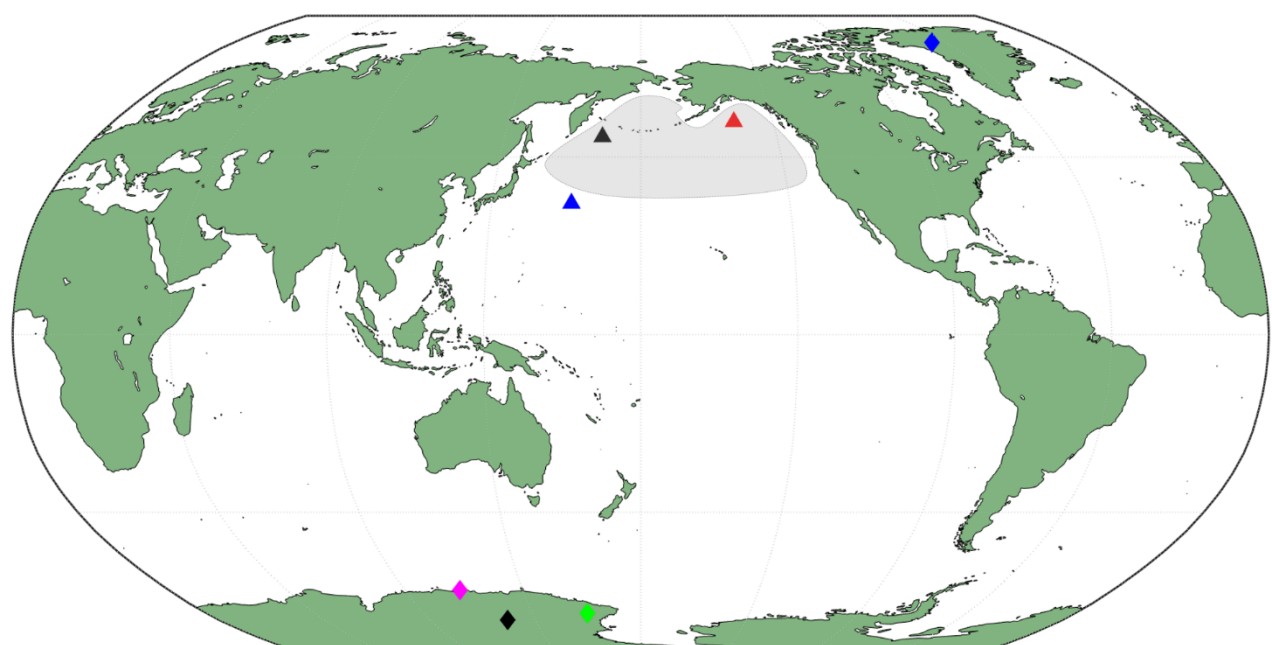










**Figure 2** – Panel a) δ¹⁸O (blue line) profile is from the NGRIP ice core (North Greenland Ice Core Project, 2007). Panel b) *nss*Ca flux (red line) from the NEEM ice core. Panel c) and Fe flux (black line) from the NEEM ice core. Shaded blue rectangle: Younger Dryas. Shaded orange rectangle: Bølling-Allerød. Numbers in the upper panel indicate the Dansgaard-Oeschger events from 3 to 16.

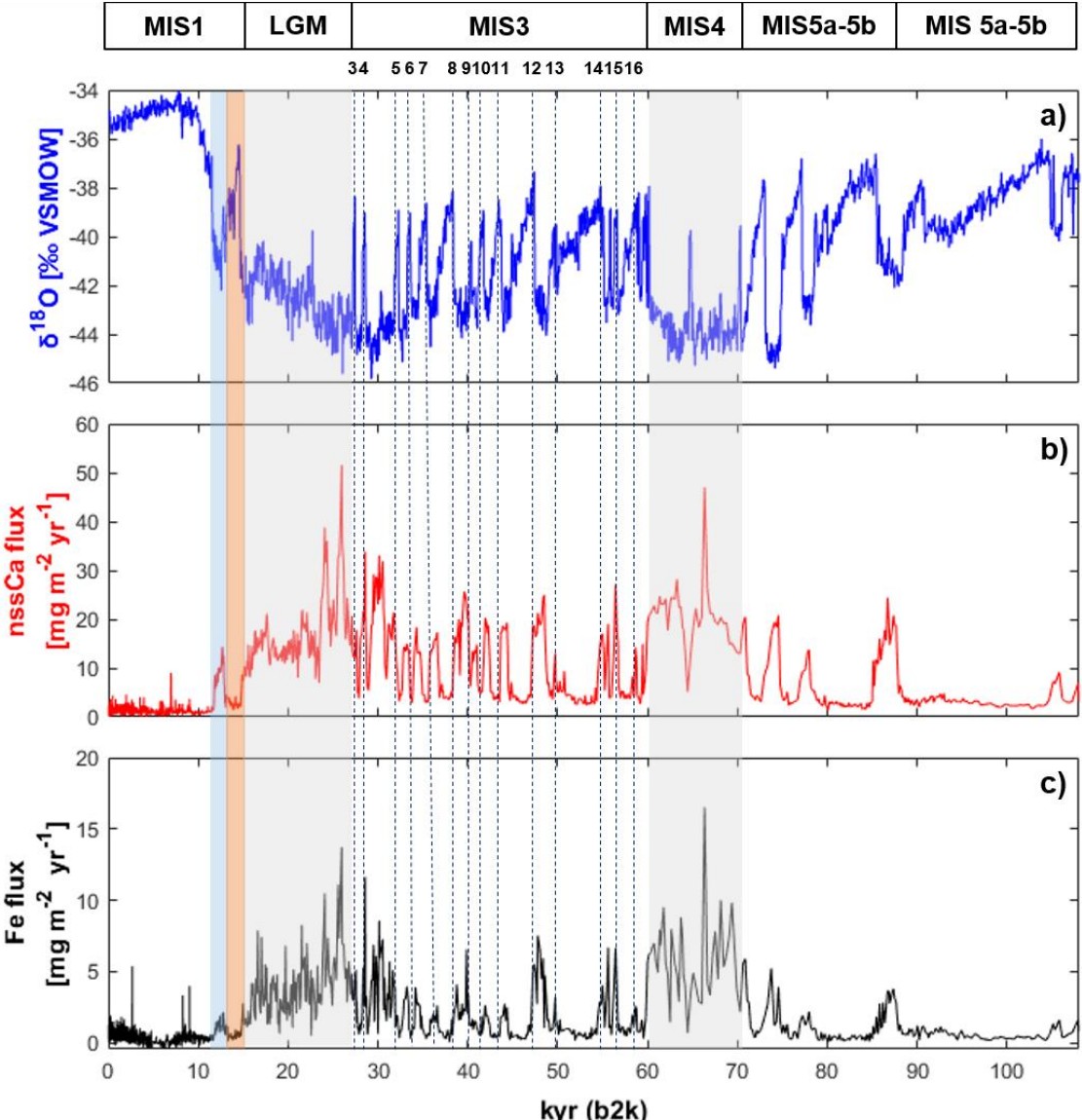

**Figure 3 –** Comparison of the Fe fluxes among a) NEEM (this work, pink diamond), b) TD (Vallelonga et al.,
2013; green diamond) and c) EDC (Wolff et al., 2006; black diamond). Note that the y-axis for NEEM ranges
from 0 to 20 mg m$^{-2}$ yr$^{-1}$, while the y-axis for TD and EDC ranges from 0 to 2 mg m$^{-2}$ yr$^{-1}$.


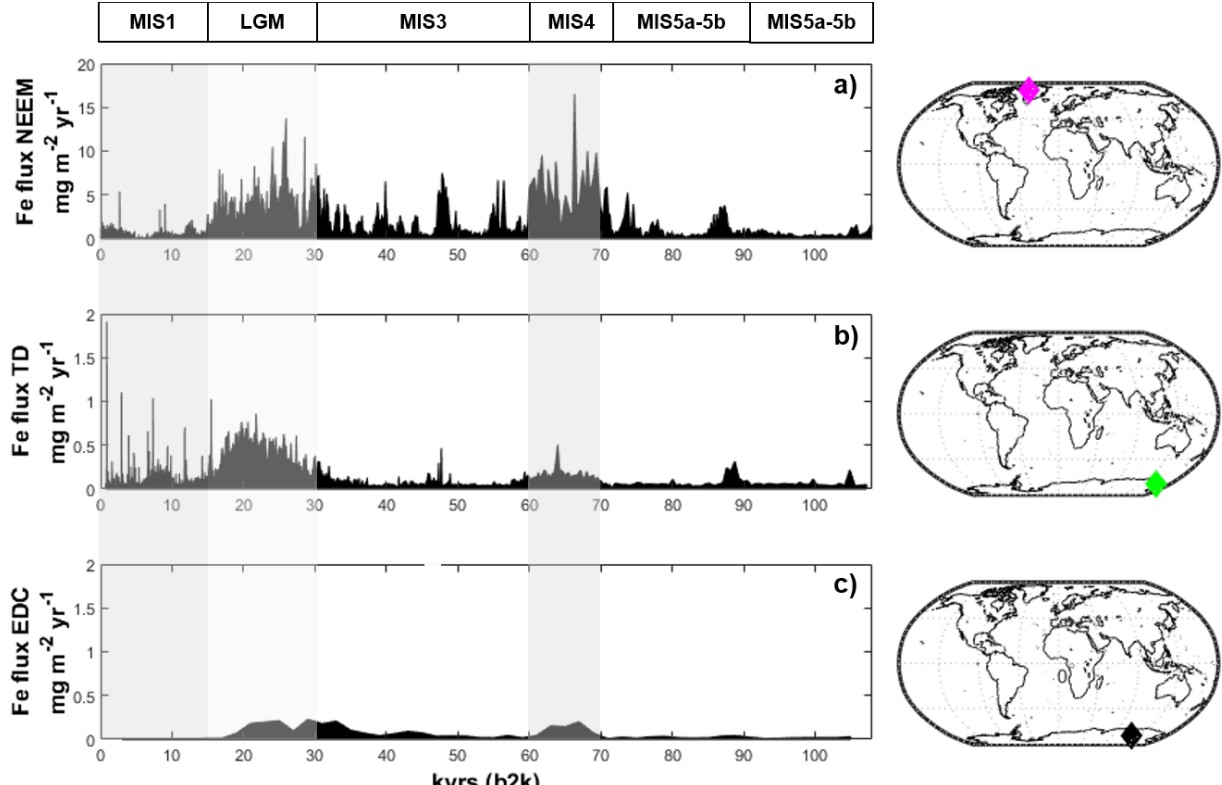















**Figure 4** – Comparison between Fe fluxes (black line, panel a) from NEEM (this work; pink diamond), with
marine productivity (red line, panel b) from ODP887, eastern subarctic Pacific (McDonald et al., 1999; green
triangle), ODP882 (red line, panel c), western subarctic Pacific (Haug et al., 1995; black triangle) and S-2 (red
line, panel d), mid-latitude North Pacific (Amo and Minagawa, 2003; red triangle). Due to their limited
temporal extension, productivity records from SO202-07-6 and SO202-07-26 are not discussed in this figure,
but in Figure 4.

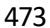


**Figure 5** – Relationship between Fe flux in the NEEM core, and MPP in subarctic Pacific Ocean over the last
26 kyr, where higher brassicasterol-total organic carbon ratio represents an increase in productivity. Sea-ice
data are from Meheust et al. (2018): prevalently extended sea-ice (dark blue rectangle), prevalently marginal
sea-ice (blue rectangle), prevalently variable sea-ice (light blue rectangle), prevalently ice-free (white
rectangle). Fe flux record (black line, panel a), productivity in the eastern subarctic Pacific Ocean (SO202-07-
6, red line, panel b) and productivity in the western subarctic Pacific Ocean (S0202-27-6, red line, panel c).
Productivity pulses were recorded when sea-ice changed its conditions towards ice-free conditions. YD =
Younger Dryas, B/A = Bolling-Allerod event, HS1 = Heinrich Stadial 1, LGM = Last Glacial Maximum.

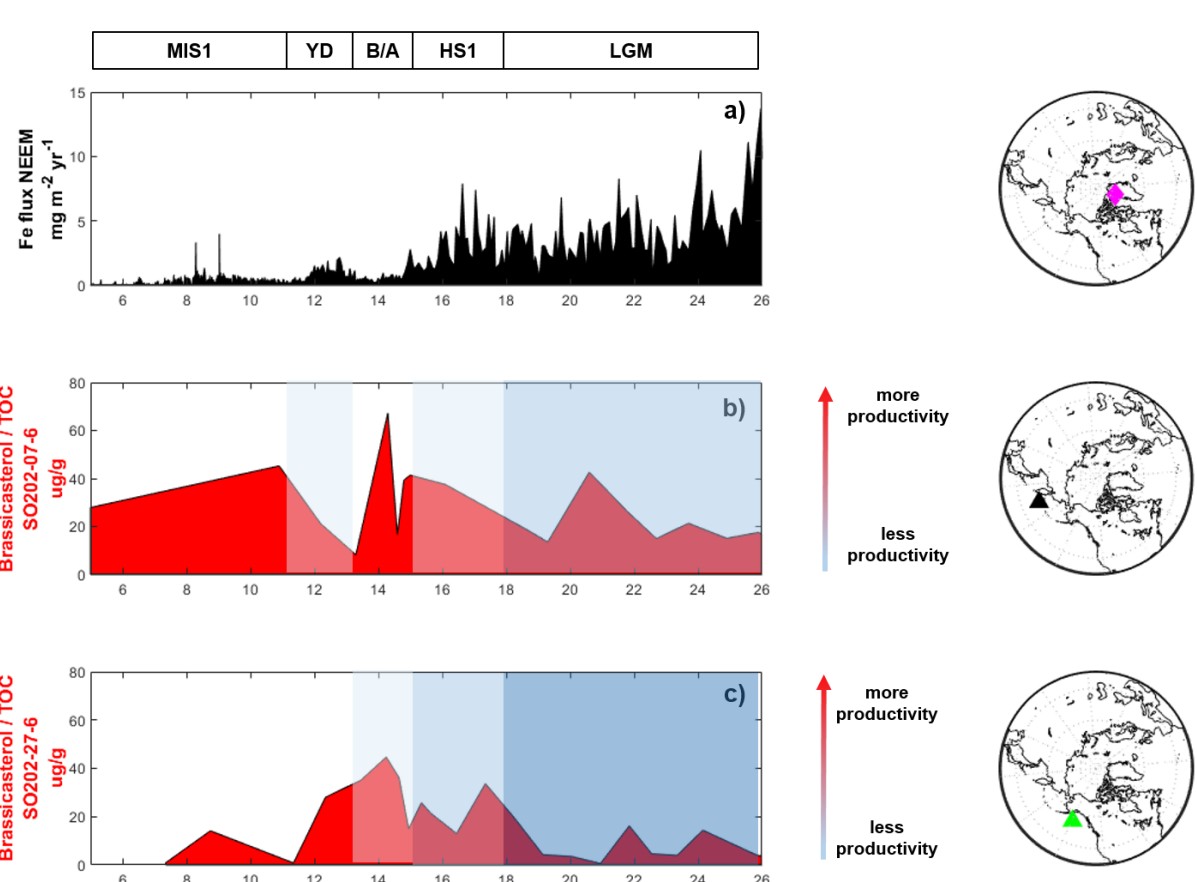

**Table 1 -** Temporal resolution of NEEM ice core, accordingly with the GICC05modelext-NEEM-1 age scale
(Rasmussen et al., 2013). Ice samples for ICP-MS analysis were collected with a resolution of 110 cm.

| Temporal resolution | Period |
|---|---|
| 10 years | Holocene (present-7.2 kyr) |
| 22 years | Holocene (7.2 kyr-LGM) |
| 110 years | Last Glacial Maximum |
| 73 years | Interstadials |
| 147 years | 28-59 kyr |
| 440 years | 59-70 kyr |
| 220 years | 70-96 kyr |
| 730 years | 96-110 kyr |




**Table 2 –** Fe and nssCa average concentration (ng g$^{-1}$) and fluxes (mg m$^{-2}$ yr$^{-1}$) from the NEEM ice core. More
details in the text. The coefficient of variability (CV) was calculated for Fe and nssCa fluxes and it is reported
in bold.

| | Fe average concentration /ng g$^{-1}$ | Fe average fluxes /mg m$^{-2}$ yr$^{-1}$ | nssCa average concentration /ng g$^{-1}$ | nssCa average fluxes /mg m$^{-2}$ yr$^{-1}$ |
|---|---|---|---|---|
| **Holocene** | 2.9 | 0.5 | 7.2 | 1.4 |
| *(0.042 -11.7 kyr b2k)* | | **(CV 1.2)** | | **(CV 2.3)** |
| **Glacial** | 44.3 | 2.0 | 210.8 | 10.0 |
| *(11.7– 108 kyr b2k)* | | **(CV 1.1)** | | **(CV 0.8)** |
| **Younger Dryas** | 18.2 | 1.2 | 135.2 | 8.5 |
| *(11.7 – 12.9 kyr b2k)* | | **(CV 0.3)** | | **(CV 0.4)** |
| **LGM** | 86.3 | 3.6 | 273.3 | 12.3 |
| *(14.5 – 26.5 kyr b2k)* | | **(CV 0.6)** | | **(CV 0.7)** |
| **MIS 3** | 45.5 | 1.9 | 216.6 | 10.2 |
| *(26.5 – 60 kyr b2k)* | | **(CV 1.0)** | | **(CV 0.8)** |

| | [Fe] | Fe flux | [Fe] | Fe flux |
|---|---|---|---|---|
| **MIS 4** | 146.4 | 5.8 | 510.2 | 20.5 |
| *(60 - 71 kyr b2k)* | | **(CV 0.5)** | | **(CV 0.3)** |
| **MIS 5a-MIS 5b** | 17.0 | 1.1 | 98.6 | 6.3 |
| *(71-87 kyr b2k)* | | **(CV 1.0)** | | **(CV 0.8)** |
| **MIS 5c-MIS 5d** | 6.5 | 0.8 | 50.4 | 4.3 |
| *(87-108 kyr b2k)* | | **(CV 0.8)** | | **(CV 0.9)** |






**Table 3** – Comparison of average Fe concentration ([Fe] in ng g$^{-1}$) and fluxes (in mg m$^{-2}$ yr$^{-1}$) among four different ice cores: NEEM, Talos Dome (Vallelonga et al., 2013), Law Dome (Edwards et al., 2006) and Dome C (Wolff et al., 2006). n.a. = not available. Average Fe concentration at DC is not available since the accumulation rate at that site during MIS4 is unavailable. Data from Law Dome spans from 59 to 8.5 b2k (for the Holocene) and from 18.2 to 23.7 b2k (for the LGM). The coefficient of variability (CV) was calculated for Fe fluxes and it is reported in bold for all the cores.


| | Greenland | | Antarctica | | | | | |
|---|---|---|---|---|---|---|---|---|
| | **NEEM** | | **Talos Dome** | | **Law Dome** | | **Dome C** | |
| | **[Fe]** /ng g$^{-1}$ | **Fe flux** /mg m$^{-2}$ yr$^{-1}$ | **[Fe]** /ng g$^{-1}$ | **Fe flux** /mg m$^{-2}$ yr$^{-1}$ | **[Fe]** /ng g$^{-1}$ | **Fe flux** /mg m$^{-2}$ yr$^{-1}$ | **[Fe]** /ng g$^{-1}$ | **Fe flux** /mg m$^{-2}$ yr$^{-1}$ |
| **Holocene** *(0.042 -11.7 kyr b2k)* | 2.9 | 0.5 **(CV 1.2)** | 1.4 | 0.09 **(CV 1.2)** | 0.09 | 0.04 **(CV 0.5)** | 0.2 | 0.007 **(CV 0.2)** |
| **LGM** *(14.5 -26.5 kyr b2k)* | 86.3 | 3.6 **(CV 0.6)** | 10.3 | 0.4 **(CV 0.5)** | 2.4 | 0.4 **(CV 0.7)** | 16 | 0.15 **(CV 0.5)** |
| **MIS4** *(60- 71 kyr b2k)* | 146.4 | 5.8 **(CV 0.5)** | 3.1 | 0.17 **(CV 0.4)** | n.a. | n.a. | n.a. | 0.12 **(CV 0.6)** |

| | | | | | | | |
|---|---|---|---|---|---|---|---|
| **LGM/Holocene ratio** | 30 | 7 | 7 | 4 | 27 | 10 | 80 | 21 |
| **MIS4/LGM ratio** | 1.7 | 1.5 | 0.3 | 0.4 | n.a. | n.a. | n.a. | 0.8 |



**Table 4 –** Summary of locations and data source for all the cores (both ice and sediment cores) discussed in the text (NH = Northern Hemisphere; SH = Southern Hemisphere)

| Name | Core | Location | Reference | Latitude/Longitude |
|---|---|---|---|---|
| NEEM ice | Ice core | NH | *This work* | 77°45'N, 51°06'W |
| Talos Dome | Ice core | SH | Vallelonga et al., 2013 | 73°0'S 158°0'E |
| Law Dome | Ice core | SH | Edwards et al., 2006 | 66°46'S 112°48'E |
| Dome C | Ice core | SH | Wolff et al., 2006 | 75°06'S; 123°23' E |
| ODP882 | Marine sediment | NH | Haug et al., 1995 | 50°22'N; 167°36'E |
| ODP887 | Marine sediment | NH | McDonald et al., 1999 | 54°22'N; 148°27'W |
| SO202-27-6 | Marine sediment | NH | Meheust et al., 2018 | 54°12'N; 149°36'W |
| SO202-07-6 | Marine sediment | NH | Meheust et al., 2018 | 51°16'N; 167°42'E |
| S-2 | Marine sediment | NH | Amo and Minagawa, 2003 | 33°22'N; 159°08'E |

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
