# Peer review of "Atmospheric Fe supply and marine productivity in the Glacial North Pacific"

_Climate of the Past, 2020_

## Referee Comment (RC1) · Anonymous Referee #1 · 20 Aug 2020

General Comments: This is a concise paper providing some useful insight into linkages between iron fluxes derived from the NEEM ice core in Greenland with past marine primary production in the north Pacific Ocean. This is a nice contribution to the literature where data is sparse. However, I believe some further clarification is needed in the paper to make the results more easily understandable and convincing to a broader audience. I provide some recommendations for the authors' consideration below.

Specific Comments: L22: Specify what you mean by "our data" .. NEEM ice core? L23: The phrase "marine productivity".. how is this being defined? What proxies were used? Do you mean marine primary productivity? L25: are you referring to upwelling

[Figure]

of major nutrients here when you say "absence of upwelling nutrient supply?" L25: "Fe-fertilization" when? During the LGM? L26: "the transition zone of the North Pacific" .. unclear what you mean here.. do you mean the transition zone between and HNLC region and region that is not an HNLC region?

L37: Suggest adding in "Regardless of the source" before the sentence beginning with "The higher atmospheric dust.." L39-40: Can you expand on the implications of the effect you describe on the Earth energy budget? L43-45: Unclear what is meant by "artificial and natural" Fe-fertilization processes. Can you expand? L50 Ref for 8-20 ppmv figure? L51: What do the authors mean by "leachable Fe" ... this can mean different things depending on the fraction of Fe studied or leach used. Ultimately how bioavailable is this leachable Fe – this seems to be the key question. Did the studies cited define leachable Fe in the same way? If not, how are they comparable? L57: Seems to imply the LD site is the central plateau.. is this true? L59: What are possible sources for a homogenous load over the entire continent during the LGM.. can you expand? L65: It would be good define earlier on in the paper (such as here) what you mean by "leachable Fe" and how bioavailable or not this fraction might be?

L87-88: Were the samples filtered prior to analysis? If not, maybe consider adding "total" to the description of leachable Fe. Are you using the 30-day acidification as your definition of leachable Fe? Please state this explicitly. How representative is this of bioavailable fraction (soluble in seawater)? The authors discuss this later in the paper but this is really a central point to their linkages to impact on marine microbial production, so I think the paper's conclusions could be served by being explicit about the definition of "leachable Fe" and using previous studies to back-up why this leach choice was made. L99: Phrase "Its quantification was performed" is repetitive and unclear what it's referring too.

L137: I think it's problematic that the term "leachable Fe" has not be explicitly defined, described, or justified yet.

L144: Phrase "activation of dust deflation area" is unclear. Provide more detail? L146: Is this paragraph referencing Fig 2 still? L158: Yes, as the authors point out here "the biological relevance of Fe" is the motivation for this study so I think the biological relevance (or not) of these Fe concentrations needs to be made sooner in the paper then it currently is. L161-163: Can you expand on why this is an upper limit (given leaching conditions), why this choice of a leach was made, and what other studies have used a similar leach. Line 164-170: It's unclear to me what time period you are referring too – the past (when?) or present. Please clarify. Line 171: Are you referring to present day Fe-fertilization experiments? L164-170, 184-186: Is there a lag in time between when the E. Asian dust source influences the N. Pacific HNLC region vs Greenland that needs to be accounted for when considering primary productivity patterns? What about the influence of dust from the Sahara on Greenland? L190: Can you expand on what types of marine production these proxies estimate? L199: Expand on potential mechanisms to increase water stratification during cold period when you might expect a deeper mixed layer L220: When are you referring to when you say the region "was not characterized by stratified waters" L233: Expand on reasons for enhanced water stratification during the coldest periods? L229-240: Are you referencing Fig 2 here? Clarify? L260: I agree – and would just expand and address this a bit earlier as suggested above.

Technical Corrections L85: Two % signs typo L110: State units for C and A in ()'s. Replace "whose" with "with" L120: Add "Coincidently" before "The recorded average Fe fluxes. . ." L123: Replace sentence starting with "A" with "However," L160: Would be good to have a reference at the end referencing HNLC region of the North Pacific. In the map I suggest for Fig 1, it would be helpful to clearly show not only the location of all cores (analyzed in this study and used for comparative purposes) but also the location of North Pacific HNLC and transition zone. L164: Replace with "Previous geochemical evidence shows" L173: Replace "is rather" with "maybe" L184: Is "This location" referring to the S-2 sediment core? Unclear. L201: Phrase "determined an increase .. " is awkwardly phrased. The enhanced Fe supply determined an increase? Consider re-wording. L207: Replace "on seawater" with "on the seawater surface" L228: Clarify what you are referring to when you say "western and eastern sides". L232: Replace "similar" with "same" L235: Replace "additional effect on they phytoplankton" with "stimulation of phytoplankton production"? L239: Replace "players" with "physical processes"? L257: Replace "This indicates" with "This may indicate"

Figures It would be helpful if the first figure was a map of the different sites where the data was collected and being compared too.

Figure 2: Does the axis on the 3rd panel have a typo .. should it be EDC? Or maybe just be consistent with calling it EPICA Dome C (DC) as you do in text? Can you highlight the time periods you are discussing in text on figure?

---

## Referee Comment (RC2) · Anonymous Referee #2 · 2 Sep 2020

In this manuscript, Burgay et al. present a new Fe record from the NEEM ice core in Greenland, and compare the observed Fe fluxes to ocean sediment records of productivity to examine potential links between Fe deposition and marine primary production in the Northern Pacific. The main takeaways from this work are that Arctic Fe fluxes are comparable-though with a few differences-to previously published ice core dust records (nssCa from Greenland as well as other Fe records from Antarctica), and that there is not a strong link between NEEM Fe flux and marine primary production in the North Pacific HNLC region, except for in the transition zone.

Major comments: Overall, the short format of this manuscript is appropriate for the

scope of the interpretation. I think that the introduction would benefit from including a paragraph better describing "leachable iron" (compared to other types of Fe measured in ice) and implications for bioavailability. This is mentioned later in the text, but is a crucial aspect of the interpretation that I do not think is sufficiently addressed and should be mentioned earlier in the manuscript- have other studies of ice estimated what fraction of leachable Fe is bioavailable or if soluble and total leachable Fe covary?

I also think the comparison to Antarctic Fe records could be improved. Does the measurement resolution and methods for each record permit a valid comparison? The authors note that fluxes are lower at Law Dome and Dome C- it seems that the samples from Dome C (Wolff et al., 2006) may have only been acidified for 24 hrs prior to analysis- would that lead to under-recovery of Fe? Furthermore, can the comparison between Arctic and Antarctic records be used to say anything about dust source regions? Through the fluxes may be comparable between the two poles, dust source regions are very different.

The comparison to the marine sediment cores could be strengthened with some additional context and information. What is the rationale for choosing the specific marine sediment cores to compare to in this study? What is meant by the "transition zone"? Can the HNLC and Transition zones be included on one of the maps? I would also hesitate to make statements as strong as the ones on lines 243-244 saying that "the transition zone of the N Pacific was sensitive to atmospheric Fe supply" and that "a direct link between Fe transport and ocean productivity holds only in the transition zone of the North Pacific" (lines 254-255) solely based on comparisons to one marine sediment core record and without a better understanding of how much of the ice core Fe is bioavailable.

Additionally, a literature search returns a recently accepted manuscript at another journal that also presents Glacial-Interglacial Fe measurements from the NEEM ice core. I think it would be necessary to acknowledge this and include relevant discussion comparing to this record/interpretations during revisions: Cunde Xiao, Zhiheng Du, Mike

[Figure]

J Handley, Paul A Mayewski, Junji Cao, Simon Schüpbach, Tong Zhang, Jean-Robert Petit, Chuanjin Li, Yeongcheol Han, Yuefang Li, Jiawen Ren, Iron in the NEEM ice core relative to Asian loess records over the last glacial-interglacial cycle, National Science Review, nwaa144, https://doi.org/10.1093/nsr/nwaa144

Lastly, the English of this manuscript could be improved to allow the reader to better interpret/assimilate findings- some of the sentences and phrasing are complex and difficult to understand throughout.

Other comments:

Lines 42-44: Sentence is confusing- consider rephrasing.

Lines 77-78: What is meant by a "low-resolution sampling apparatus"? Is this just manual collection?

Lines 79: I would replace "not contaminated" with "least likely to be contaminated" unless there is evidence for zero contamination.

Line 85: Two "%" symbols

Lines 88-89: Is there a citation to support dissolution of particles after 30 days?

Lines 105-106: Were any replicate samples run to assess reproducibility?

Line 107: Change header to "Results and Discussion"

Line 114: Include dates for Holocene to be consistent with other discussion

Line 144: What is a "dust deflation area"?

Lines 169-170: Provide some more context for these statements. Is this during modern times or from paleo studies?

Line 173: "Rather" is colloquial

Line 178: Define "high-resolution"

[Figure]

Lines 195-204: This paragraph needs to be rephrased to say that nitrate is d15N is used to interpret stratification in the first sentence. This is not clear until the last sentence, making the entire paragraph seem out of place

Line 213: What is the "transition zone"?

Line 228: "Figure 2" seems to be the wrong figure for this statement.

---

## Referee Comment (RC3) · Anonymous Referee #3 · 24 Sep 2020

Review of "Atmospheric Fe supply has a negligible role in promoting marine productivity in the Glacial North Pacific Ocean" by Burgay et al. This study presents the dust and iron flux (sourced from dust) in the NEEM ice core from MIS5c-5d to MIS 1. These records are compared to East Antarctic ice core records (Talos Dome and Dome C) to observe the similarities or differences between hemispheres. The authors find some similarities between Talos Dome and NEEM (interior East Antarctic site has lower dust and Fe flux), but found considerable differences during MIS 4. More text expanding on why this difference is observed, and why these specific ice cores were chosen for comparison would benefit the manuscript. Given the potential role of dust-borne Fe on marine biological activity, the authors then correlate their ice core dust and Fe record

to marine sediment core records of biological productivity. Authors find a correlation between dust-borne Fe transport and biological productivity in the transition zone of the North Pacific but not in the subarctic Pacific. However, the records of Fe supply to these specific sites are not presented (only proxies for biological productivity), so they may be receiving different amounts of dust flux and Fe composition compared to what is observed at NEEM ice core. It would be beneficial for the authors to acknowledge this assumption. That being said, the title of the manuscript may need to be altered. I have outlined specific suggestions for improving the manuscript below.

Specific comments:

The authors should comment on changes in water stratification that may occur during glacial periods in the North Pacific, is productivity is more limited by nitrate rather than Fe? What are the upwelling conditions like near the sediment core records you are comparing the ice core dust flux to? Need more information that suggests atmospherically transported Fe (versus upwelling) is the primary source of Fe to these sites.

Can you comment about oxygen content in the Glacial North Pacific during this time period?

Need to define the time period covered by the Holocene in the text, currently only in Table 3? Make sure all time periods are defined in the manuscript.

Why don't you compare your NEEM record to other Greenland ice core records? Or more information is needed as to why you chose Talos Dome, Law Dome, and Dome C specifically to compare your NEEM Fe flux to.

Abstract: Please list the time period that the data covers in this study.

Throughout the manuscript, past tense (e.g. was explained) is used instead of present (is explained, or is attributable to. . .). I suggest changing to present, but this is a stylistic request that is up to the authors.

lines 16-17: One sentence is attributing aeolian dust as one of the main Fe sources

to the ocean and the second sentence is stating that ice cores provide a sensitive and continuous archive for reconstructing Fe fluxes over last millennia. I suggest the authors add a sentence or portion of a sentence stating how aeolian dust transported over past climate periods is preserved in the ice core record.

Line 19: In a portion of the Arctic region over the last 108 kyr.

Line 21: remove "the" before Marine Isotope Stage 4.

Line 21: Avoid starting sentences with "They", instead combine the two sentences together or redefine what "they" is.

Line 22: Comparison of our data with. . .

Line 23: "we found that the coldest periods are characterized by the highest Fe fluxes, but marine productivity in the subarctic Pacific Ocean did not increase likely due to greater sea-ice extent and the absence. . ."

Line 29: ". . .that provide records of temperature, atmospheric dust load, and atmospheric gas composition variability during the Holocene and late Pleistocene (refs)"

Line 31: There are many ice core records that provide this information besides what is cited here, I would suggest being specific about the ice cores that were presented by the studies cited here, or expand the references listed.

Line 31: Glacial periods were dustier and characterized by a lower $CO_2$. . ..

Line 32-33: This dichotomy is explained through several different hypotheses: the increase in aridity and newly exposed continental shelves. . .

Line 34: What do you mean by "enhancement"? Be more specific here.

Line 34: an increase in the aerosol. . .

Line 35-36: increased glacial-derived mobilization of highly bioavailable iron (Fe) from physical breakdown of bedrock. . .

Line 36-37: more vigorous polar circulation capable of entraining addition dust from lower latitudes.

Line 38: through both physical and biological mechanisms. Dust particles can absorb and scatter incoming solar radiation and outgoing infrared radiation...

Line 40: Conversely, once deposited on the ocean surface, the mineral dust delivered major and micronutrients (including Fe).

Define "iron" as "Fe" and use this throughout the manuscript

Line 43-44: should also cite Shoenfelt et al., 2017; High particulate iron(II) content in glacially sourced dusts enhances productivity of a model diatom, Science Advances 3, e1700314.

Line 53: the average flux and concentration values of dust?? or leachable Fe?

Lines 62-63: the evolution of global atmospheric circulation

Lines 66: which provides unique insight

L 67: and the Last Glacial Period (suggest choosing a format for this and sticking to it throughout the entire manuscript)

L 68: "and various palaeoproductivity records from the..." Which records? list them here.

L 95: A 120 second rinsing step with 2% HNO3 occurred after each sample analysis to reduce any possible memory effect, and the vials used...

L98: ...it was quantified using the interference-free isotope 57Fe and external calibration curves with acidified standards...

L116: The last 4000 years are characterized by...

L137: What makes this fraction the leachable Fe concentration? Are you assuming that the 1 month leaching at a pH of 1 in HNO3 is the labile portion?

L144:. . .is likely related to the proximity of the ice core site to nearby regional dust deflation areas in Victoria Land that may not reach the central Antarctic Plateau (Delmonte et al., 2013)

L146: The LGM (19- 26.5 kyr b2k) was characterized by Fe fluxes on the same order. . .

L152-153: . . .compared to Antarctica.

L159:. . .one important question remains regarding whether its increase in flux triggered the marine productivity. . .

L160-163: This is an important point. To know the truly labile portion of Fe present in the ice core dust, it would be necessary to leach the dust in conditions similar to what is observed in the modern ocean (with the assumption that the pH and chemistry of the modern ocean is similar to what was observed during MIS 4 and the LGM). Could be useful to discuss that a bit more here.

L164: "Geochemical evidence indicates the dust source influencing Greenland and the North Pacific is similar in origin from the East Asian deserts (references)." Is this in the ice core record or marine sediment record?

L164: What time periods? glacial and interglacial? What about Lupker et al. 2010 [Isotopic tracing (Sr, Nd, U and Hf) of continental and marine aerosols in an 18th century section of the Dye-3 ice core (Greenland), Earth and Planetary Science Letters 295, 277-286] who suggested Sahara as an additional potential source?

L166: ". . .is primarily deposited over the HNLC region. . ."

L168: ". . .may reflect potential Fe fertilization effects promoted by increased atmospheric Fe supply."

L170: "resulted in enhanced MPP by more than 60% in this region."

L171-173: would suggest combining these two sentences together.

L177: For the period ranging from the LGM to the Holocene. . .

L178-181: Please include the lat, long, and depth of the sediment cores or indicate where this data is available (Table 4).

L 183-185: All sediment cores compared here share the same Asian dust source. . .

L190: need some information about what brassicasterol concentration is informing on.

L193: "The disagreement between Fe in the ice core record and MPP response may reflect the key conditions that result in intensified primary production such as well-developed water stratification. . ."

L198: replace "than during" with "compared to"

L199: which drove the system towards. . .

L201-202: second half of this sentence doesn't make sense to me.

L205-207: okay but what about when the sea ice eventually melts? How long is this sea ice thought to have persisted for? If atmospheric dust was deposited on sea ice surfaces presumably when the sea ice melted there would be a pulse of Fe to the surface ocean? It would be interesting to expand on this here.

L229: un-capitalize "aeolian"

L235: Thus, additional atmospheric Fe supply had little effect on phytoplankton productivity, suggesting their growth and productivity was likely. . .

L239: other influences (e.g. meltwater. . .)

L246: what does "This" refer to? both sentences in line 243 and 246 start with "This", try to avoid starting sentences like that.

Conclusions & future perspectives I think this section can be expanded upon, right now it just reads like a quick summary of the main points brought up in the manuscript without expanding on why we see the largest differences in the Fe record during MIS

4, what this means in terms of dust supplied Fe to subarctic Pacific in previous climate regimes (e.g. Mid Pleistocene Transition?).

L249: the first Fe record from mineral dust input?

L252: The greatest difference observed between the sites in opposite hemispheres occurred during MIS 4...

L252: What is the underlying mechanism for this large difference during MIS 4? or hypothesis? you should state that here.

L260: This study provides an upper limit for estimating the potentially bioavailable Fe supplied to marine phytoplankton, and additional studies should focus on analyzing the labile and bioavailable Fe fractions to constrain realistic Fe supply and response of the marine ecosystem.

Figures Figure 1: Suggest making x-axis in kyr rather than years. Make fonts on axes, labels, and numbers indicating Dansgaard-Oeschger events all need to be larger. The lines on all three lines is extremely faint, is it the top layer in the figure? Would be helpful to split up panels into 1a, 1b, and 1c.

Figure 2: Same comments as figure 1, all fonts need to be larger on axes labels, etc. Location of ice cores should be noted in the figure caption. E.g. pink circle indicates location of NEEM core. Would be helpful to split up panels into 2a, 2b, and 2c.

Figure 3: all fonts need to be larger, the figure should be split up into figure 3a, 3b, 3c. The shaded rectangles should be located behind the data instead of on top.

Figure 4: Font size is fine on this figure, but same suggestion about the years axes (kyr instead of years b2k). Same comment about shaded rectangles located behind data. Need to split up into 4a, 4b, and 4c. Suggest changing the introductory sentence of this figure caption, something like: "Relationship between Fe flux in the NEEM core, and MPP in the subarctic Pacific ocean over the last 26 kyr; would be helpful to give readers some context into what the brassicasterol/TOC is telling us. For example, higher ratios

indicate more productivity, and lower ratios suggest less productivity so having an arrow indicating this on the right-hand side can aid the reader in visualization.

Table 1: should the first period be: "Holocene (pre-7.2 kyr)" rather than "post-7.2 kyr"? Caption should start with "Temporal resolution of NEEM ice core" or something like that.

Table 2: need period at the end. Need to indicate that the first two are averages.

---

## Author Comment (AC1) · 27 Nov 2020

Dear Reviewer,

First, on behalf of all the authors, I would like to thank you for your precious suggestions that contributed for the overall improvement of the manuscript. Please find attached the new version. We would like to point out that in the first version there was a mistake in the calculation of the Fe and nssCa fluxes for the NEEM ice core. We repeated the calculations and the mistake is now fixed. However, this correction does not affect the interpretation of the dataset. We also included in Table 1, the nssCa concentration and fluxes since we referred to them several times in the main text.

[Figure]

We also introduced in the method sections more details about the analytical performances for Ca and Na and more indications on how nssCa was calculated. This info was missing in the first version of the manuscript (L122-131). Best regards,

Andrea Spolaor Corresponding author

Reviewer #1 We introduced a new Figure 1 and we edited ex Figure 2 as you suggested.

L23 - The phrase "marine productivity" how is this being defined? What proxies were used? Do you mean marine primary productivity? Yes, we meant "marine primary productivity". We introduced a new paragraph where the proxies used to evaluate marine primary productivity are described used at L298-L304.

L25 - are you referring to upwelling of major nutrients? Yes

L25 - "Fe-fertilization" when, during the LGM? During LGM and during MIS4. It is now specified at L27

L26 - "Transition zone of the North Pacific" We changed "transition zone" with "mid-latitude North Pacific" throughout the entire manuscript to be consistent with what reported in Amo and Minagawa, 2003.

L39-40 - Can you expand on the implications of the effect you describe on the Earth radiative budget We integrated this suggestion at L41-L45.

L51 - What do the authors mean by "leachable Fe"... Did the studies cited define leachable Fe in the same way? If not, how they are comparable? We address this critical aspect from L97 to L106 and from L167 to L171. In particular, we changed the terminology from "leachable Fe" to "Total Dissolvable Fe" as expressed in Edwards et al., 2006. Our procedure was consistent with what suggested by Koffman et al., 2014 for trace element analysis in ice cores. It differs from the procedure used for Fe analysis in TD and EDC,, meaning that absolute concentrations (and fluxes) are not directly comparable due to the different analytical procedures. However, the general

trends and features are still comparable. More details in the text. L59 - What are possible sources for a homogenous load over the entire continent during the LGM, can you expand? We now expanded this part from L179 to L185.

L65 - it would be good define earlier on the paper what do you mean by "leachable Fe" We introduced a sentence in the introduction (L75-78) that better defines what TDFe (Total Dissolvable Fe) means. We also discussed the analytical procedure at L97-106.

L87-88 - Were the samples filtered prior to analysis? Are you using the 30 day acidification as your definition of leachable Fe? How representative is this of bioavailable fraction? The samples were not filtered. Unfortunately, we cannot quantify the bioavailable fraction from TDFe, thus we assumed that it represents an "upper limit of the Aeolian Fe potentially available for the phytoplankton", accordingly with what reported by Edwards et al., 2006. However, previous studies showed a significant correlation between TDFe and DFe (i.e. likely more available for the phytoplankton), indicating that when TDFe increases, DFe increases as well. DU, Zhiheng, et al. Relationship between the 2014–2015 Holuhraun eruption and the iron record in the East GRIP snow pit. Arctic, Antarctic, and Alpine Research, 2019, 51.1: 290-298. Xiao, Cunde, et al. "Iron in the NEEM ice core relative to Asian loess records over the last glacial-interglacial cycle." National Science Review (2020). We specify the reason beyond our choice of acidifying the samples for 30 days at L95-104.

L137 - I think it's problematic that the term "leachable Fe" has not be explicitly defined, described or justified yet Now we have described and discussed "leachable Fe" at L75-78 and L97-106, we also introduced a disclaimer at L167-L171 where we underlined that, because of the different acidification times, the NEEM record cannot be directly comparable to the TD and EDC ones, even though the general trends and feature remains comparable.

L161-163 - Can you expand on why this is an upper limit, why this choice was made, and what other studies have used a similar leach? This is now reported at L97-106

and L167-L171.

L164-170, 184-186 - Is there a lag in time between when the E. Asian dust source influences the N. Pacific HNLC vs Greenland that needs to be accounted for when considering primary productivity patterns? What about the influence of dust from Sahara on Greenland? The time that dust particles spend from the E. Asian dust source to Greenland is about 10-13 days (Schupbach et al., 2018). However, some atmospheric processes during transport might occur, resulting into a different amplitude between the dust deposited over Greenland and over the HNLC North Pacific. Nevertheless, the dust fluxes between Greenland and the North Pacific sediment cores changed coherently and simultaneously during abrupt climate changes. We discussed it deeply at L265-L274. We also added a discussion about the different dust sources that can influence Greenland at L275-L285.

L190 - Can you expand on what types of marine production these proxies estimate? This is now reported at L298-L304.

L233 - Expand on reasons for enhanced water stratification during the coldest periods A deepened discussion is now reported from L297 to L306

Please also note the supplement to this comment:
https://cp.copernicus.org/preprints/cp-2020-77/cp-2020-77-AC1-supplement.pdf

---

## Author Comment (AC2) · 27 Nov 2020

Dear Reviewer, First, on behalf of all the authors, I would like to thank you for your precious suggestions that contributed for the overall improvement of the manuscript. Please find attached the new version. We would like to point out that in the first version there was a mistake in the calculation of the Fe and nssCa fluxes for the NEEM ice core. We repeated the calculations and the mistake is now fixed. However, this correction does not affect the interpretation of the dataset. We also included in Table 1, the nssCa concentration and fluxes since we referred to them several times in the main text. We also introduced in the method sections more details about the analytical performances

for Ca and Na and more indications on how nssCa was calculated. This info was missing in the first version of the manuscript (L122-131). Best regards,

Andrea Spolaor Corresponding author

We introduced a more detailed definition for "leachable Fe". On this purpose, we decided to re-name it as "Total Dissolvable Fe", following the terminology used by Edwards et al., 2006. More details are now reported at L75-78, L97-106 and L167-L171. Previous studies showed a significant correlation between TDFe and DFe (i.e. likely more available for the phytoplankton), indicating that when TDFe increases, DFe increases as well.

DU, Zhiheng, et al. Relationship between the 2014–2015 Holuhraun eruption and the iron record in the East GRIP snow pit. Arctic, Antarctic, and Alpine Research, 2019, 51.1: 290-298. Xiao, Cunde, et al. "Iron in the NEEM ice core relative to Asian loess records over the last glacial-interglacial cycle." National Science Review (2020). At L167-171 we state that a direct comparison between the NEEM record and TD and EDC cannot be done because of the different acidification times. However, the main features and general trends can be comparable. We also introduced a more detailed description of dust sources both regarding the NEEM ice core (L275-L285) and the Antarctic cores (L172-199). Long-term productivity records in the North Pacific are sparse. We focused on two regions (the eastern and western side of the North Pacific) from where we retrieved both high-resolution productivity records for the last 27kyrs (Meheust et al., 2018) and long-term productivity records (McDonald et al., 1999; Haug et al., 1995). We changed "transition zone" with "mid-latitude North Pacific" throughout the entire manuscript to be consistent with what reported in Amo and Minagawa, 2003.

L243-244: I would also hesitate to make statements as strong as the ones on lines 243-244 saying that "the transition zone of the North Pacific was sensitive to atmospheric Fe supply" and that "a direct link between Fe transport and ocean productivity holds only in the transition zone of the North Pacific (L254-255) solely based on comparisons

to one marine sediment record and without a better understanding of how much of the ice core Fe is bioavailable. We totally agree with your suggestions. We changed the sentence accordingly at L378("MPP in the mid-latitude North Pacific might have been more sensitive to the atmospheric Fe supply") and at L391 ("Merging our record with marine productivity data, we found that a link between Fe transport and ocean productivity holds in the mid-latitude North Pacific, suggesting that this area is sensitive to the atmospheric Fe supply"). For these reasons, at L397-399 we concluded that future investigations that aim to better quantify the more labile and bioavailable Fe fractions are needed to constrain realistic Fe supply and response of the marine ecosystem. Regarding the Xiao et al., 2020 paper, we added a section in the manuscript at L229-240 where we disucssed the differences between our record and their findings. We found that the different analytical procedures used might have led to different results. This highlights the need for a standardization of the trace element analysis among different laboratories to achieve reproducible and more comparable records.

L77-78: "what is meant by a "low-resolution sampling apparatus"? Is this just manual collection? Yes. We changed at L88-89.

L88-89: is there a citation to support dissolution of particles after 30 days? Yes, we reported the most relevant citation about acidification experiments: Koffman et al., 2014.

L105-L106: were any replicate samples run to assess reproducibility? Reproducibility was tested both using the TM-RAIN04 reference material reading it every 50 samples (as well as for accuracy determination). We also read six selected samples (3 from the interglacial and 3 from the glacial period) 5 times during the analytical run and we found an average RSD% of 5% (7% for samples from the interglacial periods, 4% for samples from the glacial period). Details are reported from L122-129.

L169-170: provide some more context for these statements. Is this during modern times or from paleo studies? It is referred to modern times (L253)

L213: what is the "transition zone" As stated above, we changed the term to "mid-latitude Pacific Ocean" accordingly with the terminology used by Amo and Minagawa in their paper.

Please also note the supplement to this comment:
https://cp.copernicus.org/preprints/cp-2020-77/cp-2020-77-AC2-supplement.pdf

---

## Author Comment (AC3) · 27 Nov 2020

Dear Reviewer, First, on behalf of all the authors, I would like to thank you for your precious suggestions that contributed for the overall improvement of the manuscript. Please find attached the new version. We would like to point out that in the first version there was a mistake in the calculation of the Fe and nssCa fluxes for the NEEM ice core. We repeated the calculations and the mistake is now fixed. However, this correction does not affect the interpretation of the dataset. We also included in Table 1, the nssCa concentration and fluxes since we referred to them several times in the main text. We also introduced in the method sections more details about the analytical performances

for Ca and Na and more indications on how nssCa was calculated. This info was missing in the first version of the manuscript (L122-131). Best regards,

Andrea Spolaor Corresponding author

We modified the figures accordingly to what you suggested. Your stylistic suggestions are implemented in the main text as well.

The authors should comment on changes in water stratification that may occur during glacial periods in the North Pacific, is productivity more limited by nitrate rather than Fe? What are the upwelling conditions like near the sediment core records you are comparing the ice core dust flux to? Need more information that suggests atmospherically transported Fe (versus upwelling) is the primary source of Fe to these sites. We discussed more in details the reason behind the stronger water stratification during the Last Glacial Period. During that time, the nitrate consumption efficiency was high, despite the low marine primary productivity suggesting an iron limited primary production. More details at L309-327. Considering the negligible role that aeolian Fe fertilization had in these regions during glacials, we assume that other Fe sources played a more relevant role in regulating MPP in the subarctic Pacific Ocean. More details at L347-L356.

Can you comment about oxygen content in the Glacial North Pacific during this time period? We briefly commented about the oxygen content in the glacial North Pacific when we discussed the causes that enhanced water stratification during the Last Glacial Period (L314-319).

Why don't you compare your NEEM record to other Greenland ice core records? Or more information is needed as to why you chose TD, LD and EDC specifically to compare your NEEM Fe flux to. We added a new section where we discussed differences and similarities between our record and a lower-temporal resolution Fe record from the same location. More details at L229-L247. There are not other Fe long-term records for the Arctic region. To our knowledge, iron records from TD, LD and EDC are the only

one that cover a time period which is comparable to the NEEM record.

L16-17: one sentence is attributing Aeolian dust as one of the main Fe sources to the ocean and the second sentence is stating that ice cores provide a sensitive and continuous archive for reconstructing Fe fluxes over last millennia. I suggest the authors add a sentence or portion of a sentence stating how Aeolian dust transported over past climate periods is preserved in the ice core record. We modified accordingly (L15-L18)

L137 - What makes this fraction the leachable Fe concentration? Are you assuming that the 1 month leaching at a pH 1 in HNO3 is the labile portion? We changed the terminology from "leachable Fe" to "Total Dissolvable Fe", accordingly with Edwards et al., 2006. This fraction represents the amount of Fe that can be effectively dissolved from mineral particles at pH 1 for one month. Our acidic digestion procedure was made following well established protocols as described in Koffman et al., 2014. More details, references etc... are reported at L75-78, L97-106 and L167-L171.

L160-163: this is an important point. To know the truly labile portion of Fe present in the ice core dust, it would be necessary to leach the dust in conditions similar to what is observed in the modern ocean. Could be useful to discuss that a bit more here. As for the previous answer, we clearly stated that this represents an upper limit to the amount of Fe that can actually be available for the phytoplankton. We underlinde that previous studies have reported a correlation between DFe (Dissolved Fe) and TDFe (Total Dissolved Fe), meaning that periods with higher TDFe were also periods with higher DFe: DU, Zhiheng, et al. Relationship between the 2014–2015 Holuhraun eruption and the iron record in the East GRIP snow pit. Arctic, Antarctic, and Alpine Research, 2019, 51.1: 290-298. Xiao, Cunde, et al. "Iron in the NEEM ice core relative to Asian loess records over the last glacial-interglacial cycle." National Science Review (2020). L164 - What time periods? Glacial and interglacial? What about Lupker et al, 2010 who suggested Sahara as an additional potential source? We discussed the possibility of other dust sources at L275-L285 where we also reported findings from Han et al., 2018 which refer directly to the NEEM ice core.

L190 - Need more information about what Brassicasterol concentration is informing on We added a paragraph that discusses more in details the proxies used for the determination of past marine productivity (L298-L304)

L205-207 - okay but what about when the sea ice eventually melts? How long is this sea ice though to have persisted for? If atmospheric dust was deposited on sea ice surfaces presumably when the sea ice melted there would be a pulse of Fe to the surface ocean? It would be interesting to expand on this here. Unfortunately, we do not know for how long sea-ice persisted in the investigated regions. However, we know that during seasonal and marginal sea-ice conditions, productivity was higher than during perennial sea-ice periods, which suggests that when sea-ice melted it might have provided micronutrients (as well as sunlight) to the ocean system (L334-336). Conclusions & future perspectives - I think this section can be expanded upon, right now it just reads like a quick summary of the main points brought up in the manuscript without expanding on why we see the largest differences in the Fe record during MIS 4, what this means in terms of dust supplied Fe to subarctic Pacific in previous climate regimes (e.g. Mid Pleistocene Transition_)

L252 - What is the underlying mechanism for this large difference during MIS 4? Or hypothesis? We added a more comprehensive and detailed explanation on the possible reasons behind the enhancement of Fe transport during MIS 4 in Greenland at L210-228. For this reason, we did not discuss it further in the conclusions keeping them short and essential.

Please also note the supplement to this comment:
https://cp.copernicus.org/preprints/cp-2020-77/cp-2020-77-AC3-supplement.pdf

---

## Author Response (AR2)

Dear editor and reviewer,

Thank you very much for your precious suggestions that contributed to the overall improvement of the manuscript.

In the following our answers to your comments. We also made other small adjustments (all marked in red).

Best regards

Francois Burgay, Andrea Spolaor & co-authors

**Editor's comments**

*Tile: I recommend changing the title again…. the phrase "might have had a negligible role…" is not very satisfying in a title. You could consider something more general like "Atmospheric Fe supply and marine productivity in the Glacial North Pacific Ocean" or "Aeolian Fe flux in the NEEM ice core and potential linkages to North Pacific productivity during the last glacial cycle"*

We believe that your suggestion "Atmospheric Fe supply and marine productivity in the Glacial North Pacific Ocean" fits and we changed accordingly.

*Line 14 (and elsewhere): when defining acronyms, the words only get uppercase letters if they are proper names or formal terms. Eg. high-nutrient low-chlorophyll (HNLC)*

We changed also Marine Primary Production, Last Glacial Period, Total Dissolvable Fe and other words (with lowercase letters)

*Line 16, 68: "over the last millennia" implies a much more recent record…. Suggest revising*

We changed into "the last glacial cycle", also considering that this manuscript covers the last glacial cycle.

*Line 18: Suggest "….reconstructing past aeolian Fe fluxes."*

Done

*Lines 20-23: Suggest revising so that you provide some actual Fe fluxes rather than just saying they were lower or higher during particular periods.*

We introduced in brackets the average Fe flux value.

*Line 21: "Fe fluxes at the NEEM site were four times lower…."*

Done

*Lines 34/35: Could use a citation here.*

We added two citations: Lambert et al., 2008 (already cited in the main text) and Luthi et al., High-resolution carbon dioxide concentration record 650.000-800.000 years before present. Nature 453 (2008)

*Line 189: "...most relevant dust source was southern South America…."*

Done

*Line 233: "...from the NEEM ice core was recently published…."*

Done

*Lines 237-243: Suggest breaking these sentences up a little more so that you can be more specific/clear about the differences between your data and the Xiao et al 2020 NEEM Fe data. In the ideal case, it would be possible to plot the two datasets together so that readers can directly compare.*

We rewrote the paragraph introducing the differences as a bullet point list. We believe that in this way is more easily readable and the differences are showed in a clearer way.

*Lines 278-288: This is a pretty general point and maybe not worth getting into, but given the NEEM location I'm surprised earlier workers on dust provenance didn't consider the Canadian Shield or Arctic Archipelago as potential source regions. On the timescales in question, Laurentide and Innuitian ice sheets meltwater systems would've generated plenty of available sediment for aeolian transport to the nearby NEEM site. It's possible I'm mischaracterizing the previous work... have only really looked at the Han 2018 paper. Anyways, this isn't really directed at your paper in particular but just a point to consider for dust provenance.*

We agree and we added a short sentence saying that additional investigations are needed to better identify the dust sources that influence Greenland (L279-280)

*Line 289: Do you mean something like "All variables considered…"?*

Yes. Changed accordingly.

*Line 325: "Propose" instead of "report" hypotheses?*

Done

*Line 389: Given the Xiao paper, suggest "In this study, we provide a high-temporal-resolution Fe record from …."*

Done
* * *
**Reviewer's comments**

*This is a revised manuscript which I believe has adequately addressed the points raised by previous reviewers. The manuscript should be carefully read to ensure that there are no grammatical mistakes, a few are listed below.*

*The questions regarding stronger water stratification during the Last Glacial Period are addressed clearly in the revised manuscript, with more explanation for how the authors came to this conclusion for the readers.*

*Authors added more information regarding the leaching procedure that was used to determine the labile portion of Fe, including acidifying samples for 1 month with 2% nitric acid, consistent with Koffman et al., 2014. They also highlight the importance for maintaining the same procedure for other ice core trace elemental studies for adequate comparison which I fully agree with.*

*Minor comments*

*Suggest changing title of section 2.1 to: "Sampling and processing"*

*Line 133: don't start sentence with the abbreviated name*

*Line 134: replace "stays for seawater" with "indicates seawater".*

*Line 139: replace "ans" with "and"*

*Line 140-141: replace "whose values are from" with "previously determined by…"*

*Line 213: Add "During" at beginning of sentence*

*Line 231: replace "scopes" with "scope"*

*Line 245: replace "consistently" with "consistent"*

*Line 277: remove the "…"*

*Line 279: "Strontium and lead isotopes indicate that Saharan dust contributed to the overall NEEM dust budget primarily during…"*

All these comments were implemented in the main text and marked in red.

*Line 282-285: rewrite this sentence, its confusing as written now*

We rewrote the sentence (L274-276)